# A Review on the Stability, Sustainability, Storage and Rejuvenation of Aerobic Granular Sludge for Wastewater Treatment

K. S. Shameem and P. C. Sabumon *

School of Civil Engineering, Vellore Institute of Technology, Chennai Campus, Chennai 600127, India
* Correspondence: pcsabumon@vit.ac.in; Tel.: +91-44-3993-1048; Fax: +91-44-3993-2555

**Abstract:** Aerobic granular sludge (AGS) is a recent innovative technology and is considered a forthcoming biological process for sustainable wastewater treatment. AGS is composed of the dense microbial consortium of aerobic, anaerobic, and facultative types of bacteria. The mechanism of AGS formation and its stability for long-term operation is still a subject of current research. On the other hand, AGS makes the treatment process sustainable in a cost-effective way. However, in order for AGS to be applied in a broader range of applications, there are several challenges to overcome, such as slow-speed granulation and the disintegration of AGS after granulation. Many factors play a role in the stability of granules. The storage of granules and the later use of them for granulation startup is a feasible method for reducing the time for granulation and maintaining stability. This review focuses on the granulation process and characteristics of AGS, granulation time and the stability of AGS under different conditions, the comparison of different storage methods of granules, and their recovery and rejuvenation. From this review, it is evident that additional research is required to assess the effectiveness of regenerated AGS after prolonged storage to promote AGS technology for commercial applications.

**Keywords:** aerobic granulation; stability; granule storage; recovery; reactivation; sustainability

## 1. Introduction

As a novel microbial community, aerobic granular sludge (AGS) is capable of eliminating nitrogen, carbon, and phosphorus, as well as other pollutants simultaneously [1,2]. AGS is regarded as being one of the most effective wastewater treatment technologies because of its high biomass retention, lower energy usage and footprint, and good settling [2–5]. AGS technology has been implemented in the full-scale treatment of municipal and industrial wastewater [5].

The discovery of granular sludge in anaerobic blanket systems was first reported in the 1980s [6]. Afterward, in the 1990s, carrier-less AGS was developed in sequential batch reactors (SBRs). Wastewater treatment technologies such as "Nereda" are substituted globally, for example, there are many mainstream full-scale systems using AGS technology for treating urban and industrial wastewater [5,6]. Doubtlessly, the AGS technology has a lot of advantages when compared with the conventional activated sludge process (ASP) [5,6]. Conventional ASP takes a longer time to treat wastewater and it requires a large land area to build the plant [6]. On the other hand, AGS technology requires less hydraulic retention time compared to ASP, resulting in a very compact reactor with a 75% reduced size. Moreover, other proven benefits include reduced wastewater treatment costs and energy requirements [7]. Because of these advantages, the scientific and engineering communities have become increasingly interested in AGS technology [5,8]. For example, scientists employ AGS technology for the full-scale treatment of highly toxic and refractory substances in wastewater, such as nitrogen, particulates, phenol, heavy metals, pharmaceutical compounds including antibiotics, and so on.

Sequencing batch reactors (SBRs) are a widely used wastewater treatment process worldwide and AGS is commonly grown in SBRs during wastewater treatment [7]. When compared to other wastewater treatment methods, only a single tank with a lesser area is needed for SBRs. SBRs have a defined cycle of operation that lasts for 3–6 h [7,9]. Each cycle of operation is further broken down into a filling period, a mixing or aeration period, a settling period, and an effluent drawing or decanting period [7]. Among the main features of SBRs are its operation cycle and time periods, which play a direct role in the formation of granules [7,10].

In full-scale reactors, the challenge of forming granular sludge from flocculent sludge is emphasized and also more time is required to start the granulation process [11]. The inoculation of reactors using granules that were formed in enduring treatment plants is one of the most intriguing options for reducing start-up time and expanding the applicability of AGS technology [11]. However, transporting AGS in wet form from one location to another may be time-consuming and expensive. As a result, questions concerning appropriate biomass storage methods and its reactivation become pertinent. Because of the variable circumstances over which granular sludge is dependent when being stored, the granule reactivation period after storage becomes unpredictable [11]. In this regard, difficulties must be overcome during granular storage on various substrates in order to retain microbial activity, structural stability, and integrity [11]. Several other characteristics must be properly explored before adopting the strategy of using stored and pre-formed AGS as an alternative for the start-up of the AGS in another reactor. Furthermore, the comparison and feasibility of various storage systems need to be understood thoroughly. Currently, there is no literature comparing different storage mechanisms and their reactivation times except for one study which is related to the review of storage and re-activation of AGS [12]. However, this study lacks a detailed analysis and comparison of different storage methods, the performance after reactivation, and its effectiveness as well as rejuvenation. In this context, the present review focuses on the granulation process and characteristics of AGS, granulation time, and stability of AGS under different conditions which include operational conditions, substrates used, types of inoculum sludge, and physical and chemical parameters leading to AGS disintegration. Additionally, the comparison of different storage methods of granules, their recovery, rejuvenation, and sustainability aspects of AGS are discussed. The work reviewed helps to advance the AGS technology by promoting more research in the storage and re-use of granules for sustainable wastewater treatment.

## 2. Aerobic Granulation Process

Aerobic granulation is the result of the interaction between the sludge particles and the microorganisms present in the sludge and this interaction will result in small and compact spherical-shaped aerobic granules [13]. In the matrix of extracellular polymeric substances, the microbial cells are self-immobilized. The granulation is favored by the environmental conditions, operations, and feeding, short settling time, larger height-to-diameter ratios, and stronger hydrodynamic shear forces in the SBR [14]. Granule formation is a gradual process that begins with seed sludge, mainly in the form of flocs, and progresses to microbial aggregates by the action of hydrodynamic forces, thermodynamic forces, gravity, diffusion of particles, Van der Waals forces, ion-exchange inter-particulate binding, and cell membrane fusion in later stage and then to granular sludge by synthesizing more and more EPS (Extracellular Polymeric Substances). This promotes the granulation process, and finally, mature granules are formed. This is illustrated in Figure 1. AGS can be cultured using inoculum in lab-scale reactors using wastewater prepared synthetically and also on a pilot scale with real wastewater [14]. The AGS developed in laboratory-scale SBR is more stable than that of pilot-scale reactors [15]. These granules are made of different layers which are described in Figure 2. The outermost zone of this granule is the aerobic layer since it is exposed to the outer environment. This layer consists of aerobic types of microorganisms. The middle or the second zone is the anoxic layer, and the innermost layer

is the anaerobic layer, which consists of anaerobic microbes due to the scarcity of oxygen in depth.

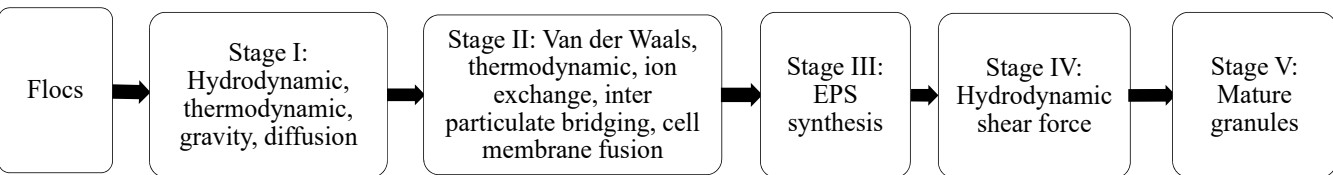

**Figure 1.** Stages of granulation.

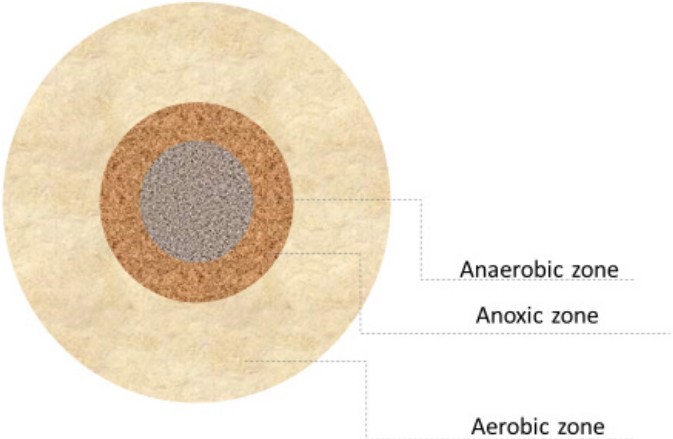

**Figure 2.** Schematic diagram of different layers of AGS.

The simultaneous removal of phosphorus, nitrates, nitrites, and ammonia can be accomplished within one reactor itself when large-size AGS is used. This is because, within a single granule, electron donors and acceptors diffuse, creating different redox conditions [15]. This aids in the growth of several microbial guilds within the same granule. High phosphorus removal is possible due to aerobic and anaerobic environments of granular structure as well as the SBR operation mode. Enhanced biological phosphorus removal (EBPR) and phosphorus precipitation within the granular matrix are the main two pathways for phosphorus removal in AGS systems [16]. In EBPR, polyphosphate accumulating organisms in the AGS will release phosphorus to the liquid in the anaerobic phase, then utilize phosphorus from the liquid in the aerobic phase, storing it as intra-cellular polyphosphates within their cells and using it for energy production. Assimilation, simultaneous nitrification and denitrification, and anaerobic ammonium oxidation (anammox) are some of the nitrogen removal processes used in AGS. All of these processes work well to remove nitrogen from the wastewater; it has been reported that AGS-based treatment can remove as much as 78.4% of total nitrogen and 99.7% of $NH_4^+$-N [17]. Microbes responsible for nitrification predominate in the aerobic zone, whereas heterotrophic microbes found in the anoxic zone are primarily involved in denitrification. Assimilation is another way of removing nitrogen in addition to nitrification and denitrification. Nitrogen will be assimilated by the generation of new biomass at the time of the granular development phase.

In an aerobic granule, soluble proteins such as enzymes cannot be secreted into the main liquid due to the EPS matrix that accumulates surrounding the cells. The polysaccharide complex, which forms the core of the EPS matrix, may capture the proteins that are released by the cells [17]. Because of this, the proteins secreted into the bulk liquid may be reduced by the EPS matrix and the granular structure of the AGS. Thus, if the activated sludge process is substituted by the AGS technique, superior effluent quality with reduced organic-N might be anticipated. In a significant study, it is clearly articulated that, the DO of the bulk solution was limited due to the sCOD and $NH_4^+$-N depletion during the initial

period of the aeration phase, creating an anaerobic core within the aerobic granules that promoted denitrification [18]. Afterwards, DO levels increased until attaining a saturation point, resulting in increased oxygen transfer within the granules. Moreover, the COD to N ratio was about 3 at that point, which was insufficient for the complete denitrification process. Enhancing denitrification must be one of the main purposes of AGS technology in order to guarantee low nitrate levels in the effluent. It can be attained by increasing the COD/N ratio and maintaining low DO and redox conditions. Volatile fatty acids (VAFs) in wastewater are easily accessible carbon sources for microorganisms. More complex carbon sources are typically converted to VFAs before they are used by microbes. VFAs act as electron ($e^-$) donors during denitrification, with nitrate serving as the terminal $e^-$ acceptor [17]. The concentration of VFAs in municipal wastewater can range from 22 to 91.6 mg-COD/L [19]. This quantity is typically insufficient for effective nutrient removal in SBR operation. So, the addition of VFAs to wastewater is recommended for improved biological nutrient removal (BNR) processes and it is possible by recirculating sludge hydrolyzed effluent from the sludge treatment section during the anoxic phase of SBR operation.

## 3. Characteristics of AGS

The stability of granules may depend on one or more attributes of the granules. Better knowledge and understanding of the characterization and formation of granules would help in knowing more about the stability of granules. The characteristics of aerobic granules can be broadly classified by their physical, chemical, and biological attributes and are described below.

### 3.1. Physical Characteristics

The physical as well as chemical constraints in the reactor contribute to the strength of the granulation system. The physical characteristics of AGS which depend upon the stability of AGS include settling properties, specific gravity, density, and physical strength. The settling ability is one of the main important factors that seem to be related to solid–liquid separation and biomass retention capacity [20]. The settling velocity of the AGS ranged from 20–80 m/h and sometimes more than 100 m/h, which was more than that of flocs with a range of 7–10 m/h [21]. This high settling velocity indirectly promotes granule stability by enabling desirable biomass retention in the reactor. Physical strength is the next important parameter that determines the capability and the structural trait to withstand high shear force and abrasion. A denser biofilm develops as shear stress increases and these shear forces regulate granule outgrowth and become more significant as the microbial population grows faster. Hence, it is vital to maintain a proper balance between shear and growth in order to encourage the formation of dense, stable granules. The reactor type and its operating parameters determine the granule's exposure to outer shear forces. The strength of the granules is a function of the interaction of certain forces that act in the reactor, the concentration of EPS, and wastewater. The granular physical strength has been measured and compared using a number of methods, including measurement in a stirred flask, mixing in a stirred type of tank reactor with a conventional system layout, and in a bubble column that can withstand high shear force and abrasion. Smaller-sized particles are likely to be denser than large-sized particles. So, these smaller-sized particles tend to have more stability than large-sized ones [22,23]. The physical strength of bacterial granules varies from that of fungi granules. Bacterial granules are stronger than fungi due to the loose macro structure of fungi which may be broken easily [21].

The operation and design of AGS depend heavily on granular density. In veritably, the two types of criteria collectively affect granule density [24]. The buoyant granule density, also known as the wet granule density, is the first parameter to consider [24]. It is the granular mass including all components present in it (other microbial cells, water, precipitates, EPS, etc.) per unit of volume of the sample. The granule biomass density, also known as dry granule density, is the second factor referred to as granule density and is defined as the mass of total dry solids per granule volume [24]. The buoyant density or wet

density of granules has been reported to be between 1005 and 1070 kg/m$^3$ [24]. A granule's wet density is related to the fluid dynamics of the AGS reactor. A force balance can be used to estimate a granule's settling velocity based on its size and density, and thus, wet density is a critical characteristic for hydro-dynamic design and for the stability of granules in the reactor [24]. The dry density has been reported to be in the 50–100 gVSS/L range [24,25]. The dry density affects the diffusion of the substrate into granules because, with more cells and EPS, the diffusing molecules are more severely constrained [24]. The pycnometer method, settling velocity method, the Percoll density gradient method, and the dextran-blue method are the different methods for measuring granular density. Among these, the pycnometer is the most reliable method and is efficient with less standard deviation and easy to use. Density directly relates to specific gravity. AGS formed in SBRs had a range of specific gravity of 1.004–1.1 kg/m$^3$, which was much more than that of flocculent sludge (1.002–1.006 kg/m$^3$) [17]. Density and specific gravity can be increased by adding some carrier materials or precipitating materials. For instance, calcium-rich granules tend to have more compressive strength and specific gravity than less calcium-containing granules [26]. Overall, the granules should have higher physical strength, settling properties, density, and specific gravity in terms of stability. The granules that are denser and have higher specific gravity tend to have higher settling properties making the granules stable. The majority of research related to physical strength is mainly focused on granular settling velocity and other settling characteristics. Limited works in literature are available based on the density of granules, which directly affects the settling characteristics of granules. The aspects of granular shear stress, hydrodynamic shear force, compression property, and viscosity need to be further explored in the physical attributes of AGS.

*3.2. Chemical Characteristics*

Aerobic granules are primarily composed of six major elements: C, H, O, N, P, and S [27]. Granule samples are dried until constant weight at 105 °C and then pulverized to analyze the elemental compositions. Then, elemental components of granules can be determined using a CHNS/O analyzer, and multi-elemental analysis can be performed using an inductively coupled plasma (ICP) emission spectrometer. In addition to that, numerous analytical methods, including X-ray diffraction, chemical extractions, scanning electron microscopy, and energy dispersive X-ray analysis (SEM-EDX), can be used to examine AGS to identify chemical deposition and precipitation, such as calcium precipitation inside the granules. The chemical characteristics include hydrophobicity, pH, and EPS. Surface hydrophobicity has a dominant function in the granulation process. The hydrophobicity of AGS is double when compared to that of bio-flocs [28]. Based on thermodynamic theory, the rise in the cell hydrophobicity could lead to a depletion of the surplus Gibbs energy in the cell and thus promote cell-to-cell self-agglomeration. Numerous analytical techniques, including contact angle measurement, bacterial adhesion to hydrocarbons, phenanthrene adsorption, and angle measurements, can be used to determine the cell hydrophobicity of AGS [29,30].

The physical and chemical properties of sludge are affected by pH, which is also a key factor in the development of biological processes. pH also controls bacterial metabolism, alters the nature and contents of EPS, and influences the pollutant removal capabilities and structural stability of sludge granules. Despite being able to maintain their structure and settleability in acidic to neutral pH environments (pH from 5.5 to 7), AGS had a negative irreversible effect on their stability in alkaline environments [31]. Likewise, strong acid (pH less than 3) may hydrolyze EPS, causing the AGS to become unstable and disintegrate. The stability of granules under different pH levels is explained in Section 4.6.

EPS of various types play distinct roles in the formation of AGS which include loosely bound EPS (LB-EPS) and tightly bound EPS (TB-EPS). The influence of LB-EPS on sludge changed to repulsion from attraction during the AGS process. On the other hand, TB-EPS might facilitate hydrophobicity and zeta potential and exhibit effectiveness in the granulation process, by sludge cell adhesion and this TB-EPS is one of the main contributors

to granule formation [32]. According to the experimentation of Liu et al., when the mature granules developed, EPS concentration sustained at about 240 mg/g MLVSS, which was greater than the seed sludge (212 mg/g MLVSS), and then grew noticeably higher than 333 mg/g MLVSS [33]. The increased EPS and Sludge Volume Index (SVI) values were negatively correlated; however, the granule formation and higher EPS content were strongly positively correlated. AGS's morphology and structure are significantly influenced by EPS. On the basis of published research, the function of each matrix of EPS elements in granular stabilization has not been explained so far. The method of extraction may be one of the reasons. The compact nature of aerobic granules made it difficult to extract EPS from them.

### 3.3. Microbial Characteristics

Light microscopy, scanning electron microscopy (SEM), fluorescence in situ hybridization (FISH), and confocal laser scanning microscopy (CLSM) are employed to observe microscopically the microbial structure of aerobic granules. The content of culture media and granular structure play a key role in determining microbial diversity. There has been a report of dead cells and anaerobiosis in the center of AGS. When anaerobic microorganisms are present in the aerobic granules, organic acids and gases are produced [34]. Aerobic granules could eventually disintegrate as a result of these end products of anaerobic metabolism. A polysaccharide network surrounds the non-cellular protein core of the mature granule. At the outer regimes, cells are accumulated. A CLSM image depicts granulation as (i) cell-to-cell aggregation, (ii) the cell grows and grows to a size that hinders substrate transfer, resulting in lysis of the cell, and (iii) shear force compacts the granular structure, leaving a non-cellular polysaccharide (PS) and protein (PN) core [34]. Granules have a smooth outer surface that minimizes damage from collisions between granules, and the PN-PS core provides mechanical and physical strength to the granules. For minimizing substrate transport resistance, functional strains are mainly distributed at the rim regime [34,35].

The specific oxygen utilization rate (SOUR) is the measurement of microbial activity. It is reported that aerobic granules have 34–168 mg $O_2$/g VSS/h of SOUR values. The SOUR is associated with increased superficial velocity, which is typically a result of liquid shear. Bio flocculation and the stability of its microbial flocs are directly dependent upon the surface charge of the cell. As the PN to PS ratio increases, the surface negative charge of the microbe is decreased, thereby lowering microbial electrostatic repulsion and enhancing granule formation [21].

The microbial communities in flocs and granules have been the subject of numerous research, but generally speaking, they are more comparable with granules and flocs in the same reactor than between similar fractions in separate reactors [36]. However, in the future it is possible that when the AGS technology matures to handle complex wastewaters, microbial diversity may be expected between granules and flocs.

### 4. Granulation Time and Stability

Granulation time and stability depends upon a number of factors which includes substrate composition, temperature, pH, organic loading rate (OLR), seed inoculum, etc. Granulation achieved quickly might not produce stable granules. Similarly, long-term enabled granulation may not always lead to stability. Recently, researchers have been trying to create granules with higher decontamination efficiency that are stable in a short period of time. The recent application of AGS and its granulation time and stability are described in Table 1. It shows that the addition of a positively charged carrier enhances the stability by binding with the negatively charged microbial surface. However, the addition of bio carriers was not improving the stability of granules and, in contrast, the granules became unstable. With the addition of polyaniline (PANI), which is a conducting polymer, the biomass retention has been improved and it also enhances the AGS pores which helped to induce polysaccharides to improve the granular stability. An investigation revealed that 2–3 mm-sized granules are more stable when compared to others by ultrasonic crushing [37]. For example, the rate of granulation of 0.0–1 mm granules significantly reduced from

92.31% to around 48.48–65.85% after ultrasonic crushing. In addition, granulation rates of 1–3 mm granules exhibited a general upward trend, peaking at 2–3 mm. However, after the ultrasonic crushing of granules, 3 mm or larger reduced slightly. The findings showed that the granules smaller than 1 mm performed poorly when it came to resisting ultrasonic crushing, whereas granules between 2–3 mm performed best in this category. Under high DO, rapid granulation is possible but has lower stability, and on the other hand, low DO requires a long start-up period but has greater stability.

Table 1. Recent application of AGS technology and its granulation time and stability.

| Seed Sludge Source | Operational Conditions | Stability Factor | Operational Period | Effect on Stability | References |
|---|---|---|---|---|---|
| Low strength wastewater | 3 reactors; Control, CCM—cell culture micro-carriers, and GAC-granular activated carbon | Slow growing organismsm and positively charged carriers | operated for 120 days | Mature granulation was attained in all reactors in less than 80 days. The particle sizes in control = 904.08 μm, CCM = 915.71 μm, and GAC = 1091.20 μm. | [2] |
| Domestic wastewater | 2 SBRs with bio-carriers | Bio-carrier | Operated for 25 days | Granulation was achieved within 20 days; However, it was not stable after 20 days. | [38] |
| Flocculent sludge from domestic wastewater treatment plant | Automated 3L SBR with 5 different phases | DO/fasting/ feeding cycle | Operated for 250 days | Granulation was observed at the end of Phase II (day 75). At the beginning of phase V, the bigger granules (1.3–8 mm) formed in the previous phase disintegrated. During phase V, stable spherical granules (1.5 to 2.5 mm) were developed. The DO (1–2 mg/L) and feeding strategy aided in the control of granule size and stability. | [39] |
| Sewage sludge | Operated in 2L with 12-h cycle and 2 cycles per day | polyaniline (PANI) addition | Operated for 33 days | The PANI injected during the start-up phase improved biomass retention, hastening the granulation process. Additionally, the PANI enhanced the AGS's pores and channels, which favored the granules' enduring stability. Additionally, PANI induced the release of polysaccharides from the cells, aiding in the maintenance of AGS stability. | [40] |
| Sewage sludge | Pilot scale SBR of volume 120.5 L, (diameter 29.2 cm, and height 180 cm) | Granular size | 40 days | After ultra-sonic crushing, 2 to 3 mm seemed to have the highest rate of granulation, the lowest SOUR was found in 2 to 3 mm granules, easier to maintain stability because their particle sizes increased at the slowest rate. | [37] |
| Brewery wastewater | 3 SBRs with 4L capacity (height = 100 cm) | Extended famine conditions | operated for 120 days with 2 different cycles of 60 days (6h cycle and 12 cycle) | Granules redeveloped with more structural stability in 12 h cycle than during the 6 h cycle period, indicating that a prolonged starving phase increased proteinaceous synthesis of EPS. Overall, prolonged famine environments favored granule stability, probably due to the bacteria with the ability to store energy molecules thrive under such settings. | [41] |
| Secondary sedimentation tank of STP | 2 SBR of 1.4 L working volume | magnetic $Fe_3O_4$@polyaniline effect ($Fe_3O_4$@PANI) | 40 days | $Fe_3O_4$ @ PANI prompted the granules for EPS secretion and better protein-to-polysaccharide ratio, improving AGS stability. | [42] |
| Anaerobic granular sludge (AnGS) and crude sewage from aeration tank | 2 L SBR | DO | 60 days | After 60 days of operation—90% AnGS turned from black color to brown color. The granules continued to be compact, and the average sizes were constant. | [43] |

A study revealed that the granular formation from anaerobic granular sludge (AnGS) as seed biomass will produce more stable granules [43]. That study claims that converting already formed AnGS into AGS with domestic wastewater rather than floccular seed sludge is one of the possible methods to cultivate AGS that reduces the long cultivation time of AGS technology, and the removal efficiency of more than 80% reveals that the granules can survive and adapt to the fluctuating concentration of the real wastewater which makes it more stable. It is observed that granule morphology changes and bacterial community changes at the phylum level during the granule development. Moreover, by comparing with the bacterial community in aerobic granules cultivated from flocculent activated sludge, some bacteria (affiliated with Comamonadaceae, Xanthomonadaceae, Rhodocyclaceae, Moraxellaceae, and Nitrosomonadaceae) playing significant roles in maintaining the structures and functions of aerobic granules. In contrast, many studies have reported that during continuous operation, aerobic granules break down. Some of the reasons for this disintegration and loss of stability are the growth of filamentous bacteria, anaerobic core hydrolyzation, functional loss of strain, and of EPS [34]. The organic loading rate (OLR) has a significant influence on the formation, stability, and structure of aerobic granules [44]. The stability and formation of AGS are discussed in detail in the next sections and Figure 3 shows the influencing factors on AGS stability.

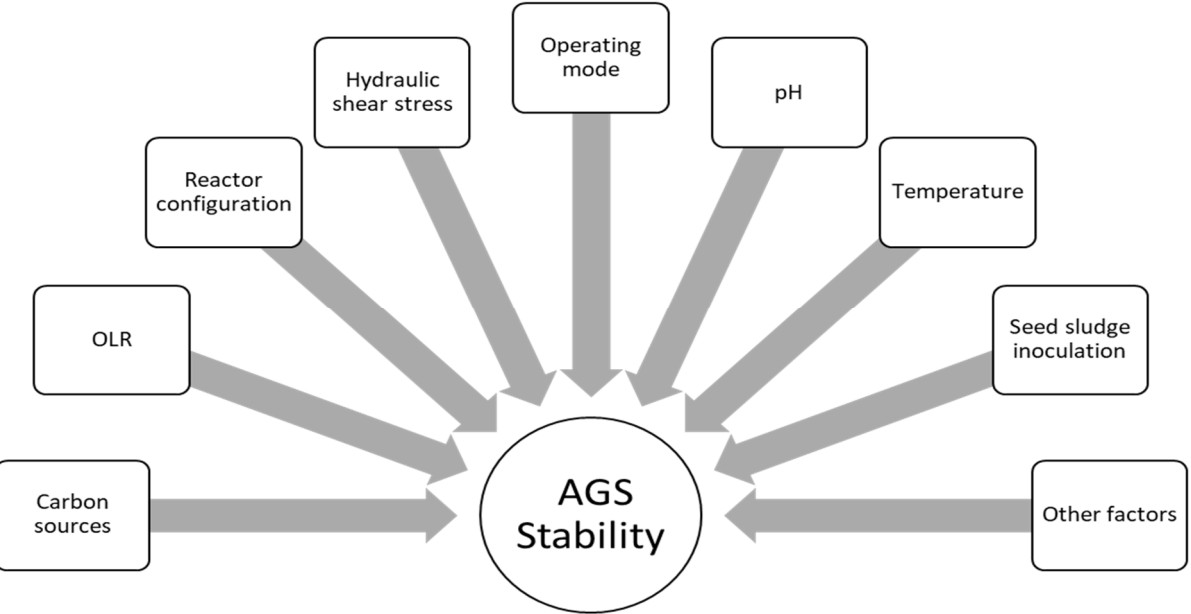

**Figure 3.** Influencing factors on AGS stability.

### 4.1. Stability under Different Carbon Sources

There are some research works that emphasize the purpose of substrate in granular formation, its maturation period, and stability. Glucose-based substrate, acetate ethanol-based substrate, and ethanol-based substrate were used for the AGS formation. Aerobic granulation was accomplished in acetate 14 days after the start-up, but ethanol and glucose exhibited granulation 40 and 60 days after, respectively [45]. Though the granulation was achieved within 14 days for acetate, the stability of granules was achieved only after two months of operation and also disintegration occurred at day 72. The disintegration probably occurred due to the formation of large diameter (>3 mm) granules, particularly closer to the reactor's bottom, and due to the thick layers formed on the biofilm made it difficult for these granules to diffuse oxygen and carbon [45]. Since there is not enough oxygen input and the carbon input is low, the anaerobic layer becomes large, thereby the most important bacterial metabolic processes in the region close to the innermost layer will be endogenous respiration and fermentation [45]. In other investigations, when acetate was used as

the carbon source, granule disintegration was noticed with long-term operation [45,46]. With that in mind, some investigations have found that the average size of the granules ought to be less than 2–3 mm, and granules developed on acetate as the substrate had the best sedimentation properties; nevertheless, when glucose is present, the AGS showed maximum resistance [45]. EPS are biopolymers made from polysaccharides (PS), proteins (PN), and other contents that act like "biological adhesives" for the development and stabilization of granules. In the EPS of stable aerobic granules, protein (PN) is the chemical that is most prevalent in EPS. Consequently, it is vital to determine the ratio between PN and PS. The only granules with a higher protein content than polysaccharides were those produced in the presence of glucose, and as a result, the impact of the carbon source on the EPS content of the AGS was amply confirmed [45]. In contrast, other experiments using glucose as a carbon source support the notion that glucose does not promote the formation of EPS [47].

The change in granule diameter (%) and stability coefficient (S) before and after the shear test are used to illustrate the resistance of aerobic granules. 'S' can therefore be used as a predictor of the stability of aerobic granules because it is connected to the stability of aerobic granules under shear stress. The detailed procedure to determine 'S' is illustrated in Nor-Anuar et al. [48] and the AGS strength increases with a decrease in the 'S' value. In other words, the more stable the aerobic granules, 'S' is an acceptable and logical way to show the AGS strength against the shear stress, even though it is not a precise tool for measuring the precise shear strength parameters in the biological process [48]. Aerobic granules are more stable for the smaller S values (percent). The acetone-fed granules had less S% when compared with glucose-fed granules. This indicates that acetone-fed granules are more stable. The microbial diversity within granules will increase the pollutant-removing capability of that AGS system. By comparing with different carbon sources, acetate encouraged the most species which gave rise to the most richness of microbes in the granules produced [45,49]. The creation and properties of the granules were influenced by the carbon source. In general, acetate facilitated the quick granule formation that produced the highest performance, and larger diameters. Though ethanol produced granules that were more stable, and it performed worse than those that were developed on acetate [45]. The substrate that required more time to granulate was glucose, and additionally, the least diverse and effective microbiotas were found in the granules grown on glucose [45]. These investigations have demonstrated the higher stability of ethanol-fed granules, but their removal and performance efficiency fell short of that of glucose-fed and acetate-fed granules. Although the microbial diversity and treatment performance were higher for the glucose-fed granules, the acetate-fed ones are more stable. Implementing them in the treatment of real wastewater is pretty challenging because the composition of the real wastewater is highly variable, despite the stability being highly dependent on these substrates feeding circumstances. Furthermore, all of these experiments used synthetic feeding in laboratory-scale reactors. In order to treat real wastewater, several investigations need to be performed in pilot-scale and full-scale reactors to assess the granular formation and its stability.

*4.2. Stability under Different Organic Loading Rates*

One of the most important factors influencing biological wastewater treatment systems is the OLR. Low OLR values (less than 2 kg COD/m$^3$ d), medium OLR values (2–4 kg COD/m$^3$ d), and higher OLR values (over 4 kg COD/m$^3$ d) are the three categories used to classify OLR levels [50]. It has been shown that AGS can occur both at extremely low OLRs (0.6 kg COD/m$^3$ d) and higher OLRs (24 kg COD/m$^3$ d) [50]. The duration of AGS formation and the nature of the granules that result are the variables. At low OLR, AGS development takes a couple of weeks, whereas at high OLR, it just takes a few hours or even day and it would be preferable for AGS to develop at a low OLR when sufficient time is available for granulation so that small sized and stable AGS are produced [50]. The fundamental issue is maintaining the stability of AGS when OLR is high because by raising

the OLR, granular size is increased, which eventually creates an anaerobic environment inside the granular core because of the restrictions on oxygen diffusion [50]. As a result, stability lowers which leads to the disintegration of granules. To sustain granule integrity, previous researchers had looked into switching between low and high OLRs. In order to enhance granule stability, a study on the combination of high OLR and feeding anaerobic is recommended. It is possible to further make use of the newly discovered technique for AG stability by OLR stressing (higher OLR for granular formation) and then reducing the OLR when a steady state condition is reached [51]. Overall, high OLR is preferred for speedy and quick granulation, while for stability and performance, an OLR of low to medium is ideal.

### 4.3. Operating Mode of Reactor

According to the available studies, SBR is typically utilized for aerobic granulation, owing to its fill-and-draw operation and stage of aerobic deprivation. This is because, the SBR's fill-and-draw mode makes aerobic granulation easier due to quick settling of sludge and SBR has two distinct periods of adequate substrate activity and substrate starvation, with aerobic starvation accounting for as much as 75% of the overall cycle. Extracellular polymeric substances are extensively consumed by microbes at the time of aerobic starvation, which lowers the sludge's surface-negative charge and increases its hydrophobicity, causing the sludge to develop dense and stable AGS [52]. Granulation in a continuous flow reactor is inherently unstable. Many researchers emphasized the importance of additional studies for granule stability in continuous flow reactors for prolonged time periods, which is highly desired [53,54]. Another reactor is an aerobic granular sludge membrane bioreactor (AGMBR), a hybrid system that involves aerobic granules as the biomass for treating, and the treated water is obtained through a membrane filter. Several literature studies are available on Membrane Bio Reactor (MBR)-based AGS systems [55]. For more than a decade, there has been a huge interest in the AGS-MBR with the outcome of the combining of AGS and MBR technologies to reduce membrane fouling. However, the key difficulty identified is in maintaining the structural stability of granular sludge, and another significant concern is the granule breaking affecting a rise in irreversible (i.e., pore-blocking) membrane fouling [56]. It is still necessary to find a solution for this issue in order to use AGS in real applications. In light of the performances under various favorable operational conditions, methods for producing more stable granules were proposed [57]. A recent literature study identified four major elements as the primary keys for developing stable AGS-MBR systems. The first one is to form granules directly in the AGS-MBR setup rather than seeding granules from SBR; the second one is to ensure the presence of anaerobic and aerobic conditions in continuous reactors; the third one is to ensure that the mean dimensions of the granules are far from the critical values (0.2 mm); and the fourth one is to regulate AGS scouring [56]. Compared to all reactors in these studies, the AGS in SBR is more stable and reliable. However, there are significant issues that need to be resolved, such as how to accomplish granule formation and ensure structural stability of AGS at the time of continuous operation of the reactor.

### 4.4. Aspect Ratio of Reactor (H/D)

In most literature studies, higher height to diameter (H/D) is preferable for stable granules. For example, SBRs with high H/D ratios took less time to reach stable granules than SBRs with low H/D ratios, even at the same operating volume [58]. Conventional wisdom holds that a height to diameter ratio of 15 to 30 is optimal for accelerating the maturation of granules with a thick surface [52]. In contrast, research has shown that granules with a dense layered structure can also develop in SBR even if the aspect ratio is relatively low [59]. This investigation has shown that when AGS properties are suitable, the aspect ratio of an SBR is not necessarily a requirement for maintaining the stability of AGS.

*4.5. Hydraulic Shear Stress*

In aerobic sludge granulation, hydraulic shear acts as a significant selection pressure [60]. AGS' structural stability is further enhanced by the hydrophobicity of its surface under hydraulic shear stress [52]. Higher hydraulic shear stress with superficial gas velocity (SGV) > 2.4 cm/s induced microbial aggregation in the granular sludge and encouraged its structural stability [60]. However, as a result of excessive hydraulic shear stresses, particles become more abrasive, which weakens their mechanical strength and incurs significant operational costs [61]. Moreover, particle structural stability and hydraulic shear stress depend on substrate concentration when it comes to sludge granulation [62]. They found that even at low SGV (0.41 m/s), wastewater with low COD concentrations (300 mg/L) produced stable granular sludge, but at 600 mg/L and higher COD concentrations, AGS could not be produced. The microbiological aggregation was enhanced, and the structural integrity of the granular sludge was favored by the higher hydraulic shear stress at SGV over 2.4 cm/s [60]. According to the hydraulic shear analysis, the SBR (airlift) with a lower aspect ratio, the overall shear rate was higher with the same SGV, whereas the total shear rate in the AGS system with a higher aspect ratio reached $(0.56-2.31) \times 10^5$/s. This showed that higher hydraulic shear stress encourages the structural integrity and development of AGS and could be provided by the reactor. Only a few literature studies are available on this aspect. Further study needs to be carried out based on this hydrodynamic shear stress to assess the structural integrity of AGS.

*4.6. pH*

Granulation is mostly stable in neutral pH. According to research findings, at pH 9, the AGS disintegrated, and the system converted to a mainly flocculent nature, leading to a considerable decrease in bioparticle dimension. The results revealed that the AGS are not stable at increased pH after a 9-day pH spike [63]. On the other hand, pH 6 shocks did not cause granular sludge breakage, and the size distribution remained constant after 9 days of operation at low pH [63]. On the contrary, acidic pH controls the epistasis of the filamentous aerobic granules and significantly alters the microbial community within the granule [64]. These filamentous microbes will affect granular stability. The stability of AGS will be impacted by these filamentous microbes. The most efficient strategy to prevent filamentous outgrowth and preserve granule stability is to keep the suspension in either a neutral or low-alkaline environment. There is no substantive study available regarding the metagenomic analysis of AGS under high or low pH. This could be tested on two systems, one of which would operate with a low pH influent and the other would operate with a standard pH. This would support the notion that the microbial ecology in a granular system is influenced by influent pH. In terms of stability, AGS is more stable in acidic to neutral pH ranges (6 to 7.3). This can be accomplished by utilizing an integrated alkali-acid pumping system in the operation of reactors.

*4.7. Temperature*

Temperature is an important factor that affects micro-organisms and their activities. The type of microbes for contaminant removal that thrive in the sludge depends on the temperature of the sludge. For instance, the ideal temperature for nitrification by Ammonium Oxidizing Bacteria (AOB) is 31 °C. On the other hand, Phosphate Accumulating Organisms (PAOs) will survive in lower temperatures. To survive, 20 to 25 °C is the ideal temperature for the majority of bacteria. The kind and growth rate of the microorganisms involved have a significant impact on AG morphology, which is related to the temperature. Temperature changes may also have an impact on AG stability. AGS will experience particle disintegration and biomass loss due to filamentous bacteria multiplying excessively at too low of a temperature, especially temperatures less than 10 °C. Whereas, the granulation system will become unstable at too higher a temperature (greater than 35 °C) due to enzyme inactivation and protein denaturation [65]. Some researchers have effectively cultivated low-temperature (<7 °C) adapted granules with excellent removal efficiency and

are highly stable so that they can expand the application of AGS in sub-freezing places [66]. Although AGS has good sedimentation capabilities at higher temperatures (30–50 °C), the effectiveness of pollutant removal declined [67]. These studies revealed that the AGS is more stable in temperatures ranging from 20 °C to 30 °C and reported the best pollutant removal performance in this range.

*4.8. Seed Sludge Inoculation*

The development of AGS was not greatly affected by the seed sludge inoculation concentration, and optimal MLSS value ranged between 1 to 20 g/L with an SVI of 7 to 220 mL/g [59]. Some bacteria in sludge are capable of strong EPS secretion, which increases the likelihood that stable AGS will form. Municipal seed sludge had a greater diversity of microbes whereas the sludge inoculation from industrial wastewater contains microbes that are resistant and adaptable to toxicity [68]. It is because industrial wastewater contains the only microbes that thrive in extreme toxicity for their survival, and they secrete more EPS which will develop stable AGS [68]. In a study, AGS was produced from saline water successfully in 52 days and effectively turned into AGS as well as retaining the structural integrity of granules [69]. It was found that the addition of stored granular sludge greatly reduces total granulation time when compared to the sludge that is flocculent, and AGS with excellent settling properties and stable structure may be created within 22 days [70]. Some researchers are focused on the addition of AGS directly into the system instead of flocculent sludge. Adding the entire granular sludge resulted in significantly reduced bulking of sludge, as well as obtaining stable AGS [71]. Overall, it is observed that industrial seed sludge is superior for stability whereas municipal seed sludge is superior for quick granulation and microbial diversity.

*4.9. Other Factors Affecting AGS Stability*

Other factors include microbial community structure, microbial quorum sensing effect, and physiochemical properties of sludge such as PN/PS ratio and particle size. The contaminant removal effectiveness and mechanical stability of AGS are significantly impacted by microbial diversity and structure. The structure of microbial communities is controlled by organic matter, pH, temperature, and stage of granulation, which are challenging to control, and cause AGS to exhibit substantial variances [72]. The structural stability of AGS will be increased by the right number of filamentous bacteria, which will aid in granulation by tying the particles to the skeleton of AGS. However, a lot of filamentous bacteria will ruin the AGS structure, which will decrease the stability. Apart from this, the stability depends on other factors which have not yet been discovered. There is scope for more studies to be carried out to find what the other factors affecting AGS stability.

## 5. Storage and Recovery of Granules

It is difficult to determine the exact mechanism of action behind the storage of aerobic granules, which is influenced by substrate, temperature, and storage time [73]. Most of the methods adopted for storage in the studies are mainly drying methods [69–71]. It has been shown that dehydrated aerobic granules, if sufficiently recovered, offer a promising long-term storage option [74–76]. In addition, the freezing and thawing method was also adopted [77]. Some studies also showed the saline storage of aerobic granules [78].

The foundation for the practicality and commercialization of AGS was to retain the stability of granules stored for a long time and to accomplish rapid recovery of physical features and microbiological activity of stored granules [74]. With prolonged storage, endogenous respiration, protein hydrolysis, and internal cell hydrolysis are thought to gradually degrade granule physical properties and bioactivity, which seems to be controlled by the substrate used, temperature, and duration of storage [75]. Biomass that was held for a lesser period of time was more resistant to breaking and had a more consistent color shift. The sections below give a brief account of storage methods for AGS.

*5.1. Freezing Method*

The majority of microbial activity will come to a halt under frozen conditions, so long-term granule storage is possible [79]. The development of granular storage at a lower temperature (20 °C) and its recovery for wastewater treatment was first reported in the year 2007 [80]. In another study, the performance of recovered granules was determined by storing the AGS at −25 °C, 4 °C, and room temperature, and recovered later [73]. According to studies it was observed that freezing temperatures (−20 to −80 °C) seemed to have no negative consequences on recovering granule's nitrification ability [81]. Nevertheless, wet conditions gradually destabilize the stored granules, partly because proteins are hydrolyzed at the cores of the granules, which weakens their integrity [82–84].

The first research that completely dehydrated and restored granules with their structural integrity was reported in the year 2018 [77]. Another study looked at the influence of varied airflow temperatures and acetone dehydration on aerobic granule drying and found that, based on their structure and reactivity, most dried granules might be retrieved to some extent [76]. A total of six different protocols were used in the study which include two air-drying methods, one freeze-drying method, ethanol/acetone dehydration methods, and one direct approach using a microwave oven [75]. They were then recovered for wastewater treatment. The measured properties are presented, and granule stability is connected with the operational stresses in the rejuvenated AGS rather than settleability, hydrophobicity, or extracellular polymeric material composition [75]. As a result of dehydration by air in the absence of moisture, minimal damage is caused to functional strains such as *Brevundimonas* and *Comamonas*, and marked deterioration is caused to structural breaker strains such as *Acinetobacter* of *Moraxellaceae*, resulting in granules that are stable and tough during rehydration. After dehydration, there is high volumetric shrinkage and color change. However, comparing this to the freeze-drying method, the shrinkage is quite low, and also the minimum color change with compact granules [75].

Comparisons of the actual granules, dehydrated ones, and the five-day recovered ones are illustrated in Table 2 [75]. Granules A, B, and E were mostly recovered after 5 days of re-cultivation, whereas D and F had deteriorated considerably, and C had rejuvenated partially and degraded partially [75]. Protocol A is dehydrated for 24 h at room temperature using adsorbent sheets [75]. The acetone protocol B involves successive immersion in different percentages of acetone (50, 80, and 100%), the ethanol protocol C, and freeze-dry technique D, which involves dehydrating for 7 h at 50 °C and then vacuum freezing for around 20 h [75]. A mean settling velocity of 12.2 m/h was determined for the original granules while settling velocity ranged from 10.8 to 17.4 m/h for the dehydrated ones. When granules were recovered after 5 days, if they were stable, they had a small increase in settling velocity. Due to the same sizes of the recovered granules, during dehydration and recovery, granule density should be raised [75]. Drying by air at 25 °C or 50 °C and dehydration protocols were used in this study to recover stable granules having a high potential for pollutant degradation [75]. Those strains having functions that remain after dehydration are enriched with recovered granules [75]. Those functional strains that remain are enhanced after dewatering with granules that are recovered and the durability and shape of granules were greatly influenced by the storage temperature [11]. COD, ammonia, and phosphorus were eliminated 13–90%, 8–27%, and 23%, respectively, using wet medium and freeze storage procedures [9]. In another work, the granular biomass was kept at 4 °C and the sample of granules was stored for 40 days to see how the storage time affected the physical and biochemical characteristics of the biomass, while the other sample was retained for 180 days [11]. In general, granules were unaffected by storage substrates, with granules at 4 °C storage exhibiting efficient recovery and granules held at ambient temperatures exhibiting the poorest restoration ability [73].

**Table 2.** Comparisons of the actual granules, dehydrated ones, and the five-day recovered ones according to Lv et al. [75].

| Samples | Settling Velocity (m/h) | | Strength-Index | | Protein (mg/g) | | Polysaccharides (mg/g) | |
|---|---|---|---|---|---|---|---|---|
| | Before Recovery | On Day 5 Recovered | Actual or Dried | On Day 5 Recovered | Before Recovery | On Day 5 Recovered | Before Recovery | On Day 5 Recovered |
| Before recovery | 12.20 | N/A | 0.0460 | N/A | 103 | N/A | 111 | N/A |
| A | 17.40 | 13.90 | 0.0230 | 0.015 | 57.4 | 34.7 | 76 | 104.6 |
| B | 14 | 22.9 | 0.0410 | 0.0028 | 63.8 | 36.2 | 74.3 | 41.7 |
| C | 15.1 | 19.3 | 0.065 | 0.002 | 28.1 | 23.5 | 173 | 38.1 |
| D | 10.8 | GD | 0.024 | GD | 90.4 | 4.5 | 59.6 | 133 |
| E | 14.4 | 18.9 | 0.025 | 0.0220 | 49.3 | 10 | 82.7 | 113 |
| F | 14.4 | GD | 0.060 | GD | 29.7 | 27.6 | 37.1 | 57.9 |

Note: GD—Granules deteriorated.

*5.2. Drying Method*

Dried sludge significantly reduces volume and weight, so the drying process facilitates the storage and transportation of granules, significantly reducing the cost of handling and shipping [9]. Moreover, unlike the freezing of dried granules, this does not require storage below freezing temperatures, saving energy costs [9,85]. It has been proposed to dry granules to facilitate storage and handling and to enhance the efficiency of bioreactor treatment by adding them as inoculums for quick start-up [86]. The drying procedure is the most practical of the several approaches recommended for preserving granules. Since it is in a dry state, it can be stored for a longer time. On re-activation, granules can be recovered in a short time with minimal loss [9]. However, previous studies have not fully developed a drying process for the dried granules to be successfully reactivated.

The original granules were yellow with an average size of 4 to 6 mm in diameter. After drying, these aerobic granules had reduced their size by half about 2.1 to 3 mm in diameter, and weight also gets reduced to 25 to 30 mg/granule [9,76,87]. The color of the granule had turned to dark brown after drying but the integrity of the granules remained. As a result of the present drying methods, granules would lose >80% of their volume and >85% of their weight [87]. However, the granules could be re-cultivated in their original state in 5 to 12 days without losing structural integrity. For this to happen, the granule matrix must be extremely resilient, capable of withstanding 80 percent volume shrinkage in drying and volume recovery in re-cultivation without breaking down; the strains must thrive in drier conditions [87]. Interestingly, the researcher has produced aerobic granules with high protein to polysaccharide ratios using seven different drying protocols: drying at 37, 60, 4 °C, in the sun, the dark, a moving air stream, or with concentrated acetone solutions and 80% volume shrinkage was experienced by all dried granules without any structural breakage [76]. Despite three recovery batches, all dried granules were restored to the majority of their original size and organic matter degradation potential, notwithstanding the loss of some volatile suspended particles [87]. In Table 3, after the first recovery, the COD removal rate of original granules was far higher than acetone-dried granules, but acetone-dried granules performed better compared to other drying methods [76]. The dark-dried, sun-dried, and air-steamed dried granules had almost the same COD removal rates [76]. The COD removal rates subsequently increased for the second recovery stage and complete COD removal was achieved within 38 h. After the third recovery, the COD removal rate for acetone-drying or dark-drying granules was even higher than that of the original granules [76]. So, some species may grow more quickly if their competitors are removed during the drying process. Nevertheless, the fact that drying can improve performance should not be taken as a blanket statement in this regard.

**Table 3.** COD removal capabilities of dried granules.

| Stage of Recovery | COD Removal Capability | Reference |
|:---:|:---:|:---:|
| **1st stage** | original >> acetone-dried > 4 °C dried > 37 °C dried > 60 °C dried > dark-dried = sun-dried = air stream-dried | |
| **2nd stage** | original > 37 °C dried = 4 °C dried > sun-dried = 60 °C dried > dark-dried > air stream-dried > acetone-dried | [76] |
| **3rd stage** | dark-dried > acetone > original > sun-dried > 37 °C dried > air stream-dried > 37 °C dried = 4 °C dried | |

*5.3. Wet Storage Method*

The wet storage method is another option for storing aerobic granules. The aerobic granular sludge is kept hydrated in this method by being kept in a moist medium. This was reported clearly that the aerobic granules are stored in tap water at room temperature (20–22 °C) for 6 months and 12 months and the COD removal efficiencies were restored to 80% of actual performance after three days of reactivation [88]. The COD elimination effectiveness increased to 90–99% after 7 days of operation, and the capacity to remove organic carbon was nearly entirely restored [88]. An ideal granule reactivation technique was proposed based on the amount of time it took to recover various operating tactics [88]. The best conditions for reactivating granules were found to be an OLR of 0.8 kg COD/m$^3$/day, a 2.6 cm/s superficial up flow air velocity, and 15–20 mg/L ammonia concentration [88]. Aerobic granules were dark brown in color after 12 months of storage and stayed elliptical or spherical with a slight size decrease, presumably due to endogenous respiration and internal cell hydrolysis of aerobic granules during long-term storage [88]. Even though, if the granules settled poorly, with 7 days of operation with SVI of 123.79 mL/g and a low settle velocity of 17.12 m/h, it is possible to resettle the granules into brown-yellow in color similar to fresh granules. Based on MLSS variation, granules gradually revived and increased steadily. Another researcher pointed out that the collected AGS in glucose solution of 400 mg/L and distilled at a temperature of 25 °C, 4 °C, and properly maintained temperature (from 20 to 26 °C) for the duration of 8 months and the findings clearly indicate that storage substrate had only a negligible impact on granules whereas temperature for storage had a significant impact on their shape and physical characteristics. The process of granular reactivation involved stabilizing their physical, chemical, and microbial characteristics. Granules held at 4 °C in glucose solution at room temperature in distilled water were ellipsoidal or spherical in shape, brown-yellow in color, thick, and transparent. When held at a temperature of 25 °C with glucose solution or water, granules were dark in color, flaky and irregular with minute holes in structure, and shape.

The process of freezing and thawing that happened when being stored causes the granules to partially break down [73]. Granules kept at 20 to 26 °C remained elliptical or spherical, with very little cell debris and granular size loss, and showed no symptoms of disintegration [73]. Granules changed color from yellow-brown to dark brown with time [73] and after reactivation, granules held at 25 °C had a great ability to settle and the PN/PS ratio remained almost unaltered, demonstrating that storage at 25 °C is better for maintaining the internal microstructure. The aerobic granules were kept in liquid media for storage at 4 °C using five storage protocols and these storage protocols are listed in Table 4.

**Table 4.** Storage protocols according to Wan et al. [89].

| Type of Media | Composition of Media | Physical Characteristics after Storage | Bioactivity after Reactivation |
|---|---|---|---|
| DI water (SW) | 500 mL DI water | Mild shrinkage, smooth surface | SOUR-28 $mgO_2/g$ MLSS/h |
| Acetone (SA) | AGS bathed in acetone solution (50–100%) for 2 h and kept in 500 mL acetone solution (100%) | No structural change, rough surface with cell agglomerations | SOUR-22.3, $mgO_2/g$ MLSS/h |
| Saline water (SS) | 500 mL NaCl solution (3% w/w) | No structural change, rough surface with cell agglomerations | SOUR-26 $mgO_2/g$ MLSS/h |
| Acetone/isoamyl acetate solution (SAA) | AGS bathed in acetone solution (50–100%) for 2 h and kept in 500 mL acetone-isoamyl acetate solution | Severe shrinkage, rough surface with cell agglomerations | SOUR-16.6 $mgO_2/g$ MLSS/h |
| Formaldehyde solution (SF) | 500 mL formaldehyde solution (10%) | No structural change, rough surface | SOUR-13.5 mg $O_2/g$ MLSS/h |

Note: DI-Deionized water.

Visual assessment of the SF, SS, and SA granules after almost a year of storage indicated no structural changes; slight shrinking of the SW granules, and extreme shrinking of the SAA granules. Isoamyl acetate miscibility after application may contribute to volume shrinkage. The SI (original granules) and SW granule displayed smooth surface, whilst some granules have rough surfaces with agglomerations of cells and this aggregation of cells can be induced by high osmotic pressure or toxic substrates and SOUR for these recovered AGS were SI > SS > SA > SAA > SW > SF [89]. All of these granules were still more active than those kept in glucose or acetate solution [90]. Apart from formaldehyde, the other stored AGS were effectively revived with appropriate performance and strength in just 24 h.

## 6. Reactivation Time of Stored Granules

Many factors influenced the storage and reactivation time of AGS, which include the storage temperature, granular morphology, storage substrate, and granular environment. There was a great deal of influence of storage temperature on granule morphological and external properties, while the substrate used for storing had a limited influence. The reactivation of AGS was actually a restoration of physical properties, the structure of granules, and microbial properties. Consequently, granules stored at 4 °C were better suited to maintain structural integrity and at −25 °C were better suited to maintain microstructure and PN/PS ratio after reactivation, which confirmed the effectiveness of storage at −25 °C in maintaining microstructure [73]. Stored AGS can be successfully employed as bio-seed for reactor fast start-up 10 days after prolonged storage, regardless of storage temperature or substrate [73].

Dried granules had a lesser reactivation period. Other storage methods indicated in the literature were not as effective as dried granules in restoring their original treatment performance. The drying methods showed higher removal rates than methods using a wet medium or freezing storage after 1d reactivation [73,88,91]. Surprisingly the recommended drying procedures could result in the effective rejuvenation of AGS in the reactor as the COD removal efficiency was 97% and no ammonia and phosphorus were detected [9]. Removal performance is stabilized or reaches the original performance after 4–16 d using a wet

medium and 1–5 d using a freezing medium, respectively [75,88,91,92]. The drying method, however, takes 1–2 d [9,76,92]. COD, ammonia, and phosphorus removal rates were 13–90%, 8–27%, and 23% for wet storage and freeze storage systems, respectively [77,91–93]. Dry storage methods, on the other hand, achieved impressive removal of 97% COD, and phosphorus and ammonia removal were nearly 100% [9]. In another work, it is reported that after 1 d of re-activation, the $PO_4$-P removal efficiency of 90%, and the performance increased and became constant till the completion of the study. On the contrary, in another study, the removal efficiency of $PO_4$-P after 1 d of reactivation was around 20%, and it took around 7 d to completely restore $PO_4$-P removal after granules have been stored in a wet media at room temperature for 12 months [88].

The wet medium and frozen granules which were stored at room temperature, 20 °C and 25 °C up to 6 to 12 months in tap water and glucose solution had a reactivation time of 5 d with $NH_4^+$-N removal rate of 5–10% after 1d reactivation and it had a recovery time of 8 to 27 d for effective ammonia removal [73]. The wet medium stored granules at 20 to 26 °C had a higher reactivation time and the organics and nutrient removal rate is lower than that of frozen ones [88]. In the drying methods, in Table 4, the stepwise acetone drying method and chemical dehydration and drying method had higher efficiency than the air-dried method. By comparing the storage methods in Table 3, the drying method after chemical dehydration showed better performance but it is stored for only 1 d. In the storage methods mentioned in Table 4, none of the methods showed the storage period exceeds 8 months. Long-term storage and the maximum duration that is possible for storing the dried granules need to be further studied.

## 7. Impact of Additives for the Establishment of Fast and Stable Granulation

Additives such as calcium ions, magnesium ions, sodium ions, ferric ions, polyaluminium chloride, magnetite nanoparticle, xanthan gum, powdered ceramsite, and calcium alginate gel beads enhanced the settleability of granules and decreased granulation period while having no effect on treatment efficiency [94]. The efficiency of treatment fluctuates, particularly in prolonged operations (more than 100 days). It is important to indicate that additives generally have no discernible impact on the removal of phosphorus, ammonia, and COD. Their major focus was on rapid granulation and repairing of disintegrated granules. The addition of nanoscale zero-valent iron (nZVI) enhanced the secretion of EPS and improved the growth rate of certain species of bacteria, including *Xanthomonadales* and *Rhizobiales*, while reducing the rate of growth of others, including *Clostridiales*, affirming that the effect of nZVI on bacterial growth was genera reliant. AGS was formed successfully in the reactors in less than 50 days, and the presence of nZVI enhanced the settling rate and size and of the granules to a certain level [95]. In a study, it was found that the addition of $Fe^0$ coated with $Mg(OH)_2$ promoted bacterial growth, improved methane concentration, and stabilized the pH in the bioreactors [96]. This can be applied to the aerobic granular sludge to obtain its benefit.

## 8. Ensuring Sustainability through the Usage of AGS

The usage of AGS is a proven method for wastewater treatment across the globe. It is easy to cultivate AGS cost-effectively and in an environmentally friendly way. Moreover, like any other treatment method, AGS does not require any media replacement, cleaning, or regeneration. In addition to this, no chemical is required for its working as well as cleaning. There is a huge scope for the usage of AGS in developing countries and underdeveloped economies around the world. This will foster the sustainable development goals of the United Nations (SPG 3—Good health and well-being, SPG 6—clean water and sanitation, SPG 11—sustainable cities and communities, SPG 15—life on land, SPG 13—climate action) [97]. As countries are currently facing economic slowdown and lacking resources, this cost-effective process using AGS may enable countries to deal with wastewater treatment effectively.

Several recent articles have highlighted the fact that AGS has a high potential for resource recovery. This includes alginate-like exopolysaccharides (ALE), tryptophan, polyhydroxyalkanoates (PHAs), phosphorus, etc., and can be extracted by different methods [98]. ALE is extracted from the extracellular matrix, and it has the capability to form a gel in the presence of uranic acid residue and calcium ions. ALE is used as a surface coating to enhance the waterproofing resistance of paper, bio-adsorbent for dye as well as in civil construction and agriculture. Tryptophan is found in EPS. The nine essential amino acids for humans include L-tryptophan. Due to its antioxidant activities, it is utilized for preserving food and as a supplement in the feed of animals. Furthermore, it has applications for pharmaceutical and agricultural purposes. Phosphorus is taken out from the wastewater in AGS systems by PAOs or accumulates in the matrix of EPS. This extracted phosphorus can be used as soil fertilizers. The PHAs are found in intracellular intrusions. PHAs are a class of biopolymers with characteristics similar to plastics that can be utilized to make paints, packaging, insecticides, strong fibers, and sanitary products, as well as having uses in biomedicine [98].

Furthermore, the storage and reuse of AGS reduce the duration of the wastewater treatment cycle. The total cost and efforts for wastewater treatment are drastically reduced through the storage and reuse of AGS. The cost of maintaining and reusing AGS is relatively low compared to other available treatment methods such as the activated sludge process, chemical precipitation, adsorption, ion exchange, etc. Some of these methods are costly, unsustainable, and cause environmental degradation and health hazards. On the other hand, the usage of AGS makes the treatment process easier, cost-effective, environmentally friendly, sustainable, and futuristic.

## 9. Research Gaps and Future Prospects

In wastewater treatment plants, it has been expected that AGS will replace the current activated sludge system. However, the granulation procedure excludes the sludge with less density and only keeps sludge that has good settling properties. This means not really being able to change overall activated sludge to AGS. Then, as a result, developing techniques to speed up the process of turning entire sludge into granules remains difficult and requires substantial study. The majority of aerobic granulation processes use SBR systems, which are the best kind of reactors. However, since the majority of treatment plants use a continuous type of reactors, implementing sequencing batch reactors in a full-scale is quite challenging to deal with larger wastewater volume.

It may take several months for AGS to form. However, more research is needed to assess the granular stability and the process of disintegration brought on by unknown variables. There is a scope for developing a basic understanding of microbial interactions and granulation. Even though efforts made to improve the AGS process's use globally, some elements of this system remain obscure. The actual use of this system and the fulfillment of its advantages are being hampered by a lack of knowledge of the underlying principles regulating the formation of biofilm, the prolonged granulation phase, and disintegration of granules when operating it for long term. Moreover, the significance of EPS and its molecular mechanism in the formation and stability of granular biofilms is unknown. In addition to that, uncertainty exists regarding the effects of various operation modes and composition of wastewater on microbial environment and EPS.

Applications of AGS in the remediation of hazardous wastes, along with heavy metal and emerging toxins adsorption are not explored much. This decontamination aspect at a higher level should be looked forward for treating more and more complex wastewater. Most of the AGS studies are based on sequencing batch reactors and stability of AGS was explored in SBR only. Till now there is no study related to the stability of AGS in continuous flow reactors. So, there is an opportunity to create a novel SGS-based continuous flow reactor design.

There is a lot of scope for developing a realistic model for the storage of AGS, pilot-scale and full-scale applications for granule storage, its recovery, and rejuvenation. Only

laboratory-scale granule storage has so far been investigated. Studies should be conducted for further investigations to evaluate the effectiveness of decontamination, removal of nutrients and organics, and granule attributes such as stability, integrity, settling, and size of recovered granules during prolonged storage. Research should be conducted to determine the viability of commercializing stored granules for enhancing the use of AGS for wastewater treatment.

## 10. Conclusions and Perspectives

Even after 25 years since the discovery of AGS, the basic mechanism of granulation is still a matter of current research. Several hypotheses have been drawn by many researchers, but it is still a subject of current research. The formation of AGS is caused by the agglomeration of several microbial guilds which form together and perform the removal of COD, nutrients, etc. It is still necessary to improve fundamental aspects of AGS in bioreactors, such as the stability of AGS and to achieve long-term and steady operation, regardless of existing knowledge in granulation. In this review, the granulation process, nutrient removal pathway and characteristics, the stability of granules, and long-term storage of granules were comprehensively summarized. It is crucial that the physical size and structure of granules be determined by the composition and operational circumstances of the feed wastewater in order for AGS systems to perform well. In fact, it is important to note that substrate type, method of influent addition; hydraulic shear forces, organic loading rate, operational hydraulic and sludge retention times, and settling time have an impact, on AGS stability. Multiple pathways contribute to the AGS disintegration, including the outgrowth of filamentous organisms, EPS hydrolysis at the substrate-depleted core, an increased growth rate of heterotrophic micro-organisms, variations in microbial distribution patterns, and heavy metal ion toxicity. This in turn decreases the density of AGS and consequently decays into small particles with very low settleability, which are easily removed from the reactor. A major focus of contemporary research in this area is to develop operational strategies for maintaining long-term stability based on a better understanding of the microstructural characteristics that ensure AG's integrity. However, the current research developments are insufficient to meet the needs of large-scale industries and inadequate for the cost-effective and efficient usage of AGS. Further investigation is required to determine granule stability and disintegration processes caused by unknown factors to support the large-scale cost-effective usage of AGS.

The storage of AGS could provide a potential remedy to the prolonged initial time frame needed to grow granular sludge and for maintaining the stability of granules. There are different storage methods for AGS out of which dehydration and drying methods are quite efficient. Due to intra-granular protein hydrolysis during storage in a wet media, the granule lost structural integrity. Granule storage at sub-freezing temperatures, however, was impractical. In comparison with wet granules, dried granules were easier to store and handle. In recent studies, most of the researchers stored granules by drying it. Dehydration and drying are preferred over other storage methods because of their effectiveness in rejuvenating and removing contaminants. Researchers need to undertake further investigations to assess the capability of AGS for recovery, after prolonged storage in terms of removal and decontamination efficiency as well as granule characteristics such as stability, integrity, settleability, and size. A new concept of commercializing stored granules for sustainable wastewater treatment at a large scale can be presented, if all these aspects are explored. Moreover, AGS has a greater potential for resource recovery such as phosphorus, ALE, PHA, and tryptophan. The storage of granules and recovering them for later use eliminates sludge disposal and its consequences. All of these factors contribute to the AGS being a more sustainable method of treatment.

**Author Contributions:** K.S.S. is involved in writing the original draft, data compilation, and improvement after each review by the senior author. P.C.S. is involved in conceptualization, data curation and writing (review & editing), project administration, and fund acquisition. All authors have read and agreed to the published version of the manuscript.

**Funding:** This research was funded by the Department of Science and Technology (DST), Government of India with grant number DST/TM/WIC/WTI/2K17/82(G4).

**Institutional Review Board Statement:** Not applicable.

**Informed Consent Statement:** Not applicable.

**Data Availability Statement:** Not deposited in any repository.

**Acknowledgments:** Authors thank VIT management in supporting the APC for publication in a distinguished open access journal.

**Conflicts of Interest:** The authors declare no conflict of interest.

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
