# Peer review of "A Review on the Stability, Sustainability, Storage and Rejuvenation of Aerobic Granular Sludge for Wastewater Treatment"

_water, doi:10.3390/w15050950_

Round 1

Reviewer 1 Report

Manuscript Title: A Review on Stability, Sustainability, Storage and Rejuvenation of Aerobic Granular Sludge for Wastewater Treatment

Manuscript Number: water-2211847-peer-review-v1

In this article, the authors have addressed the latest advances concerning the granulation process of waste sludge. The formation mechanism of aerobic granules, storage ways of aerobic granules, and operation factors affect the granulation process were comprehensively reviewed. Discussion on the main challenge of aerobic granular technology which is represented by the slow formation and the long time needed for granulation has successfully done. Also research gap and the additional research is required to advance the aerobic granular technology were well discussed.  

The writing is well-tailored, and the explanations are often reasonable. However, sometimes, at some points, the submitted research leaves something to be desired Nevertheless, the presented manuscript could be recommended for publication after a major revision and amendments following these next comments:

Comment #1: the authors need to discuss the removal mechanisms/pathways of phosphorus, ammonia, nitrate, and nitrite by the formed granules.

Comment #2: in section 2, aerobic granulation process, the authors mentioned about simultaneous removal of nitrates, nitrites, phosphorus, and ammonia within one reactor because, within a single granule, electron donors and acceptors diffuse, creating different redox conditions. But also, the authors should discus the reasons behind the elevated nitrate concentration during the operation of aerobic sequencing batch reactors and the solutions needed to efficiently treat nitrate in the aerobic granular systems.

Comment #3: we recommend the authors to discuss the chemical compositions of the formed granules based on the used techniques in the literature such as energy dispersive spectroscopy. 

Comment #4: in our point of view, the main challenge facing the application of aerobic granular technology is the start up time or the long time needed for full granulation. The authors have discussed the storage of the formed granules and the possibility of reusing them to speed up the formation of aerobic granules. As the formation of aerobic granules is governed by the growth of bacteria, there were many attempts in the literature to accelerate the formation of aerobic granules by increasing bacterial growth via the addition of trace elements in operation systems such as Iron, Magnesium, or Copper nanoparticles. Therefore, the authors can open a section to discus the effect of additives on the formation of aerobic granules and bacterial growth and the following articles are important to be cited (Impact of nZVI on the formation of aerobic granules, bacterial growth and nutrient removal using aerobic sequencing batch reactor. Environmental technology & innovation 19 (2020) 100911). (A novel method to improve methane generation from waste sludge using iron nanoparticles coated with magnesium hydroxide. Renewable and Sustainable Energy Reviews 158 (2022)112192). (Optimization modeling of nFe0/Cu-PRB design for Cr (VI) removal from groundwater, International Journal of Environmental Science and Development 12 (3), 131-138. https://doi.org/10.18178/ijesd.2021.12.5.1330). (New insight for electricity amplification in microbial fuel cells (MFCs) applying magnesium hydroxide coated iron nanoparticles. Energy Conversion and Management 249 (2021) 114877). (The impact of iron bimetallic nanoparticles on bulk microbial growth in wastewater. Journal of Water Process Engineering 40 (2021) 101825.

Comment #5: discussion on resource recovery from aerobic granular sludge process during the treating of wastewater is recommended.  

Comment #6: a comparison between the used granular technology for treating pollutants from water with other adsorbents is required in this review. The comparison shall include

the cost, performance, applicability of each method. The following articles are useful and could be cited as they investigated the removal of pollutants using different adsorbents.  (Chloramphenicol removal from water by various precursors to enhance graphene oxide–iron nanocomposites. Journal of Water Process Engineering 50 (2022) 103289). (Statistical optimization of nZVI chemical synthesis approach towards P and NO3− removal from aqueous solutions: Cost-effectiveness & parametric effects. Chemosphere 312 (2023) 137176). (Insights into boron removal from water using Mg-Al-LDH: Reaction parameters optimization & 3D-RSM modeling, Journal of Water Process Engineering 46 (2022) 102608). (Optimization modeling of nFe0/Cu-PRB design for Cr (VI) removal from groundwater, International Journal of Environmental Science and Development 12 (3), 131-138. https://doi.org/10.18178/ijesd.2021.12.5.1330).

Comment #7: Conclusions should be enhanced to cover all the study’s aspects and to highlight the novelty findings.

Comment #8: The author should consider revising the whole text formatting in the manuscript for any additional spacing, words capitalization, and unnecessary repetitions.

Comment #9: The used font type should be unified within the whole manuscript (starting with the title).

Comment #10: The use of English grammar still requires some work on the whole manuscript.

Comment #11: I would like to see the manuscript again after the revision

Reviewer 2 Report

The authors try to comprehensively summarize the granulation process and characteristics, the stability of granules and long-term storage of granules. Overall the manuscript is well-organized, clear in contents and approaches. But there are many problems in table and figure numbering and referring them in the text. Trying to be positive, the reviewer is recommending major revisions. The comments below aim to guide the authors in revision of their manuscript.

1.     Section 2-line 9: “Vander Wal force” should be replaced by “Van der Waals forces”.

2.     The writings in Fig. 1 are not so clear and it is recommended to be redrawn.

3.     Section 3.2, second paragraph, line 8: “is explained in section 0.” section 0 is wrong and must be corrected.

4.     The full form of EPS is introduced in section 2 and no need to repeat it in last paragraph of section 3.2.

5.     The full form of SVI must be introduced in the first place it’s used (section 3.2). It is introduced in section 4.8 instead of 3.2.

6.     Section 4, last paragraph: it is declared that “An investigation revealed that 2-3 mm sized granules are more stable when compared to others”. It is recommended to mention the investigated size range to be more meaningful.

7.     Section 4, last paragraph: Why the granular formation from anaerobic granular sludge as seed biomass will produce more stable granules? it should be explained in the text.

8.     There are lots of mistakes in numbering of Tables in the text and referring them. Check all and revise them please.

9.     There is a problem in starting of section 4.1 and it is cut off. What is the start of the sentence?

10.  Figure 3 is not referred in the text.

11.  What is the start of the paragraph written after Table 2? There are many mistakes like this regarding the interruption of the sentences.

12.  “drying at 37°C, 60°C, 40 °C”  should be written as “ drying at 37, 60, and 40 °C”

13.  It is common to name the figure sections as “a, b, c, ..” instead of “left, middle and right”

14.  As a general rule, all the tables and figures must be referred in the text before placing them. There are many mistakes in the text not obeying this rule! For example Table 3 is not referred in the text.

Round 2

Reviewer 1 Report

The authors have improved the manuscript well and acceptable in its present form